# Real-Time Prediction of Growth Characteristics for Individual Fruits Using Deep Learning

**DOI:** 10.3390/s22176473

**Published:** 2022-08-28

**Authors:** Takaya Hondo, Kazuki Kobayashi, Yuya Aoyagi

**Affiliations:** 1Faculty of Engineering, Shinshu University, 4-17-1, Wakasato, Nagano City 380-8553, Nagano, Japan; 2Academic Assembly, Shinshu University, 4-17-1, Wakasato, Nagano City 380-8553, Nagano, Japan; 3Research Center for Social Systems, Shinshu University, 4-17-1, Wakasato, Nagano City 380-8553, Nagano, Japan

**Keywords:** deep learning, automatic generation data, growth prediction, individual identification

## Abstract

Understanding the growth status of fruits can enable precise growth management and improve the product quality. Previous studies have rarely used deep learning to observe changes over time, and manual annotation is required to detect hidden regions of fruit. Thus, additional research is required for automatic annotation and tracking fruit changes over time. We propose a system to record the growth characteristics of individual apples in real time using Mask R-CNN. To accurately detect fruit regions hidden behind leaves and other fruits, we developed a region detection model by automatically generating 3000 composite orchard images using cropped images of leaves and fruits. The effectiveness of the proposed method was verified on a total of 1417 orchard images obtained from the monitoring system, tracking the size of fruits in the images. The mean absolute percentage error between the true value manually annotated from the images and detection value provided by the proposed method was less than 0.079, suggesting that the proposed method could extract fruit sizes in real time with high accuracy. Moreover, each prediction could capture a relative growth curve that closely matched the actual curve after approximately 150 elapsed days, even if a target fruit was partially hidden.

## 1. Introduction

Precision agriculture involves detailed observation, control, and planning of farmland and crop conditions, and substantial research on precision agriculture has been conducted to improve productivity by using information and communication technology and the internet of things to understand the growth conditions of agricultural crops [1,2,3]. Kobayashi et al. [4] developed a high-resolution image comparison system to extract crop growth information by observing the changes in their appearance. In addition, Genno et al. [5] predicted the fruit radius from apple growth curves based on the green-blue vegetation index leaf area in apple orchards. In Genno et al.’s study [5], the entire fruit tree was evaluated based on a single growth curve, while growth curves for individual fruits were not obtained. More precise management would be possible if the growth of individual fruits could be predicted.

Research on the recognition of individual fruits using deep learning to support yield mapping and harvesting robots is underway [6,7,8,9,10,11]. In addition, various attempts have been made [12] on fruit detection for workload estimation. Gongal et al. [13] developed image processing for apple identification, which can avoid repetitive counting of apples using 3D information, and achieved an accuracy of 82% for crop load estimation. However, these studies estimated the fruit area at a specific time and did not observe changes for individual fruits over time. Understanding the growth status of a single fruit would enable precise growth management and improve the quality of the products. Tracked detection of individual fruits is necessary to capture changes over time. Wenli et al. [14] were able to track the detection of oranges from dynamic point images, but they only counted the number of oranges and did not provide a follow-up growth prediction. Marini et al. [15] developed a regression model to estimate fruit weight at the time of harvest using three years of data on initial fruit diameters; however, they were unable to predict growth in real time using machine vision.

Mask R-CNN is a method for predicting regions and classes for each object [16] and has been used for strawberry [17] and blueberry detection [18]. Kaiming et al. [16] used COCO [19] as the training data for Mask R-CNN. COCO is a large dataset consisting of 200,000 annotated images with teacher labels and 120,000 images without teacher labels. Only visible regions are annotated in the COCO dataset, which makes it unsuitable for detecting hidden regions. In addition, automatic annotation methods for large datasets using different types of fruits and vegetables have been proposed [20], but automatic annotation in the case of hidden fruits has not yet been achieved.

Przemyslaw et al. [21] were able to detect hidden regions in the case of potatoes by training Mask R-CNN on a dataset that was manually annotated with regions hidden by other potatoes. However, in manual annotation, there exists a risk of learning incorrect regions, as the operator has to guess the region information. However, if a large amount of training data that can accurately determine the positional information of fruit can be generated automatically, fruit in areas hidden by leaves and branches can be detected with high accuracy.

In this paper, we propose a method for predicting the growth characteristics of individual fruits from fixed-point observation images of apples. The proposed method develops a system that automatically generates training data for deep learning, which can identify hidden fruit and enable observation of the same fruit from time-series images of apples, to identify changes in fruit size over time.

The remainder of this paper is organized as follows. Section 2 describes the training data generation method and individual fruit identification algorithm. It also describes the accuracy validation of the proposed methods. Section 3 presents the accuracy validation results of the proposed methods. The influence of automatically generated parameters of the training data and accuracy of individual fruit identification and growth predictions are also discussed. Section 4 provides concluding remarks.

## 2. Material and Methods

### 2.1. Monitoring System and Data Set

To periodically capture images from a fixed point at an apple orchard without a commercial power supply, we developed an image monitoring device that operated autonomously, based on previous research [4]. The setup of the monitoring device and its structure are shown in Figure 1. This device was powered through a solar panel (140 W, Looop Inc., Tokyo, Japan) and charged a 12 V battery (G&YU SMF31MS-850, Global Battery Co., Ltd., Korea) via a charge controller (LP-12/24V50A_2.0, Looop Inc.). Two DC–DC converters, via a charge controller, converted the electric power to 5 V and 7.4 V and supplied a microcontroller (Raspberry Pi 2 Model B, Raspberry Pi Foundation, Cambridge, UK) and a digital single-lens reflex (DSLR) camera (EOS Kiss X7, Canon Inc., Tokyo, Japan). Images captured with the DSLR camera were temporarily stored on the microcontroller and then transmitted to a web server via a USB long-term evolution (LTE) modem (L-02C, LG Electronics Inc., Busan, Korea). The microcontroller performed actions such as acquiring images with the DSLR camera and transferring them through the USB LTE modem.

Original images of apple orchards were acquired at regular intervals by the monitoring system from a fixed point. The images were RGB images with a resolution of 5184 × 3456 pixels. The existing COCO dataset and the original dataset generated from the original apple orchard images were used in this study.

### 2.2. Training Data Generation and Individual Fruit Identification System

Object detection in deep learning requires a large amount of training data, and such training using existing large-scale datasets is common. However, in fruit tree images, it is desirable to detect fruit regions hidden behind branches and leaves, but there are no existing trained models that can detect hidden regions. In this paper, we propose a method to automatically generate a large amount of training data using images obtained from the monitoring system. Songyan et al. [22] showed that data expansion is possible by applying the cut-and-paste method. In this study, the cut-and-paste method is used to automatically generate training data. The training data consist of a synthetic image of a farm and binary mask indicating the fruit area. Figure 2 shows the process of generating the synthetic images.

In the proposed method, the fruit, leaf, and background images were manually cropped and separated from the actual farm image. Fruit images were cut and separated by hand, because tools such as the Hough transform are ineffective in detecting apples [10,23]. The fruit and leaf images were then pasted at random positions on the background image to generate a composite image of the canopy. When placing the fruit and leaf images, the luminance, angle, magnification, and aspect ratio were randomly changed within an arbitrary range (Table 1) to increase the diversity of the training data. The fruit images to be placed contained randomly selected red and green apple images.

The shape was extracted from the leaf image, and an image with low luminance was composited over the fruit image as a shadow of the leaf. For the shadow images, the aspect ratio, magnification, and angle were also varied to represent shadows of various shapes. The examples of the generated training data are shown in Figure 3, and the binary mask and generated image are shown in Figure 4. From Figure 4, in the generated image, we can observe that the fruits were placed at the white positions on the binary mask.

In the generated composite image of the farm, the images of the leaves and branches were placed after the images of the fruit were placed, resulting in partially hidden fruit areas. Binary masks representing the exact fruit region were generated for each detected fruit for training using Mask-R-CNN. By generating a large amount of data combining synthetic images of the farm and binary masks, training data that could accurately identify hidden fruit areas were obtained.

The individual fruit identification system used a model trained on the proposed training data. The model extracted a contour of each fruit and rectangle surrounding the fruit from the actual farm image. The fruit size and center coordinates were calculated from the contours and rectangles. These were used to identify the same fruit at different times. The process flow of the fruit detection algorithm is shown in Figure 5.

A three-layer filter was used to detect fruit that simultaneously met the conditions shown in part ⑤ of Figure 5. The three-layer filter consisting of a distance filter, size filter, and contour filter were used to determine the same fruit at different times. The distance filter considered the hanging range of the fruit position caused by increase in weight due to fruit growth; the size filter considered size variation due to growth and depth movement; the contour filter excluded fruits with incorrect contours and non-identical fruits with similar positions and sizes by calculating the contour similarity. The distance filter extracted only the fruits for which the Euclidean distance between the center coordinates of each compared fruit was less than 40 + 2 × *T*, where *T* is the number of non-detect days. The size filter extracted only the fruits whose size variability was within 10 + 0.1 × *T* [%]. The contour filter extracted only fruits for which the contour similarity of the fruit was less than or equal to 0.05. The contour similarity was calculated using MatchShapes() in Opencv of a Python library; MatchShapes() calculates a smaller value if the similarity is greater. If no fruit satisfied the identical fruit condition, the number of consecutive non-detect days *T* was added to mitigate the limitations of the distance and size filters in the search for identical fruit on the following day. The number of consecutive non-detection days *T* was set for each fruit and reset to 0 when it was detected to be the same fruit. The above procedure was applied to the time series of farm images to obtain time series growth information for each fruit.

### 2.3. Verification of the Accuracy of Fruit Detection

To validate the accuracy of fruit detection based on deep learning, the fruit recognition accuracy was compared between a hidden fruit region detection model trained on the proposed training data and the model trained on the COCO dataset. For creating the hidden fruit region detection model, the model trained on the COCO dataset conducted transfer learning using 3000 automatically generated training datasets (1088 × 1088 [px]). Nine images (5184 × 3456 [px]), one for each day, were selected from the period 11 September to 20 November 2018, as images for validation (Figure 6). An average of 77 detectable target fruits are in the nine selected images.

As the proposed model requires 1088 × 1088 [px] images as input data, each original image that had a resolution of 5184 × 3456 [px] was cut and separated into 11 sections, generating 99 images with a resolution of 1088 × 1088 [px]. Subsequently, excluding the 6 images in which the fruit was not visible, the remaining 93 images were used as validation images. Annotation of the regions was done manually using the VGG Image Annotator (VIA). VIA can output the fruit contour in JSON format by enclosing the fruit region in the image (Figure 7).

The hidden regions of the fruit were estimated by the operator, and 1813 fruit regions were annotated. A visual inspection conducted by the operator found that 1605 of the 1813 annotated fruit regions had hidden regions.

The precision, recall, and intersection over union (IoU) were used as indices to evaluate the accuracy of fruit detection. True positives (*TP*), which indicate the number of cases where the prediction is positive and correct; false positives (*FP*), which indicate the number of cases where the prediction is positive and wrong; and false negatives (*FN*), which indicate the number of cases where the prediction is negative and wrong, are used to define the precision, as shown in Equation (1), and recall, as in Equation (2). The correct area (*CA*), which indicates the manually annotated area, and prediction area (*PA*), which indicates the predicted area of the model trained by the proposed training data, were used to define the IoU, as shown in Equation (3).


(1)
$$Precision=\frac{TP}{TP+FP}$$



(2)
$$Recall=\frac{TP}{TP+FN}$$



(3)
$$IoU=\frac{CA\cap PA}{CA\cup PA}$$


### 2.4. Verification of the Accuracy of Growth Curve Predictions

The radius of an apple fruit (*Y*) is shown in Equation (4) as the number of days *X* elapsed from April 1 in the year of detection [5].


(4)
$$Y=\frac{a}{1+b\times e^{cX}}$$


From Equation (4), a growth curve was generated in real time, and the radius at harvest was predicted according to its convergence value. When validating the accuracy of the growth curve predictions, the accuracy for radius detection and real-time prediction of the radius at harvest were confirmed. To validate the accuracy of radius detection, the automatically detected apple radius (detected value) was compared with the manually detected apple radius (true value), and the mean absolute percentage error (MAPE) and Pearson’s product ratio correlation coefficients were calculated. The true values were manually annotated with visual confirmation by the operator using VIA.

For the validation images, one image from each day in which the fruit was clearly visible was selected from the 1417 farm images recorded over 137 days from 1 July to 15 November 2016. Examples of validation images are shown in Figure 8. In the comparison, two fruit types in the farm image were targeted; the target fruits are shown in Figure 9. Fruit 1 was partially hidden by leaves and other fruit in all images, while fruit 2 was partially hidden by leaves and other fruit in 23 images, which represent several images per day ranging from 32 to 49 elapsed days.

In the validation of the real-time prediction of the fruit radius at harvest, the respective predictions obtained from the detection and true values were compared with the correct harvest radius. In the derivation of the predictions, the detection and true values at the number of elapsed days were used to identify each coefficient in Equation (3) using the least squares method, and the growth curve for each elapsed day was derived. For the derivation of the correct harvest radius, a growth curve was derived based on all true values for elapsed days 1–227, assuming 227 days to be the optimum harvest time. Each predicted value and the correct harvest radius were the convergence values of these growth curves.

## 3. Results and Discussion

### 3.1. Verifying the Accuracy of Fruit Detection

The validation results are shown in Table 2. The precision was 0.864 for the COCO learning model and 0.955 for the proposed model, with the proposed model having a higher value. The recall was 0.338 for the COCO learning model and 0.317 for the proposed model, with the proposed model having a slightly lower value. The mean IoU was 0.554 for the COCO learning model and 0.653 for the proposed model, with the proposed model having a higher value. The median IoU was 0.596 for the COCO learning model and 0.720 for the proposed model, with the proposed model having the higher value. The variance of IoU was 0.065 for the COCO learning model and 0.046 for the proposed model, with the proposed model having a lower value. The parameters of fruit height and width (Table 1) affected the precision and recall values when generating synthetic farm images. The precision decreased when the height and width were set lower than the values shown in Table 1. Setting the height and width above the values shown in Table 1 did not improve the precision and decreased the recall.

An example of a false detection is shown in Figure 10, illustrating that the COCO learning model incorrectly detected leaves as fruit, while the proposed model rarely detected them incorrectly.

In the proposed model, the recall improved and the precision and IoU decreased when small fruits were included in the training data. An example of small fruit detection is shown in Figure 11, which shows that the proposed model did not detect fruits of small sizes. The proposed model outperformed the COCO learning model in terms of the mean and median IoU, as it was able to accurately detect the hidden regions of the fruit. An example of hidden fruit detection is shown in Figure 12, which shows that the proposed model can accurately detect hidden regions caused by fruit leaves, branches, and shadows. The mean and median of IoU did not improve when the leaf size was excessively large relative to the fruit, or when the number of leaves in front was excessively greater than the number of fruits. It is, therefore, essential to set the generation parameters appropriately in order to detect the hidden regions with high accuracy. In addition, in fruit identification, if the proposed method is applied to images with denser fruit, it becomes difficult to identify the same fruit because the distance filter may not work as intended. Furthermore, previous studies on apple detection using the Hough transform [24,25] reported that the precisions were 93.5% and 92.0%, respectively, and the proposed method shows a higher value than those reported. The proposed method is effective when applied to data with large volumes and diverse state changes, such as time-series images, because the Hough transform does not detect fruit correctly with fixed parameters.

### 3.2. Verification of the Accuracy of Growth Curve Predictions

The true and detected values for each elapsed day are shown in Figure 13, which also shows the growth curves derived from the true and detected values up to 227 elapsed days. For Figure 13, in Fruit 1, the values of a, b and c in the functional approximation are a = 99.71, b = 7.848, and c = −0.024 for the detected value curve and a = 112.24, b = 6.428, and c = −0.021 for the true value curve. In Fruit 2, the functional approximations are a = 76.61, b = 8.595, and c = −0.027 for the detected value curve and a = 84.93, b = 7.001, and c = −0.024 for the true value curve. The true and detected values for the same period of elapsed days are shown in Figure 14. From Figure 14, the MAPE is shown to be 0.079 for Fruit 1 and 0.072 for Fruit 2, while the coefficient of determination for linear regression was approximately 0.96 for both Fruit 1 and Fruit 2. Examples of detections by the proposed model on 2 July and 21 October 2016, are also shown in Figure 15.

Figure 16 shows the fruit radius at harvest predicted from the true and detected values for each elapsed day. Each prediction can capture the relative growth curve that approaches the value of the true curve after approximately 150 elapsed days, even if the target is partially hidden. However, before 150 days, the predictions varied widely and had relatively large errors. This suggests that the number of samples for prediction was not sufficient up to approximately 150 days. In addition, observations from the early stages of growth are required to ensure a sufficient number of samples. Furthermore, it is difficult for the proposed method to predict the radius of harvested fruit before 150 days have elapsed, and further improvement of the growth model and prediction algorithm is required for an early prediction. Moreover, in order to avoid the large amount of processing in the study, the validation was limited to two representative fruits that were detectable over a long period of time. Therefore, although the difference in accuracy of growth prediction between the different models is not clear, it can be expected from Table 2 that the proposed model will improve prediction accuracy over the conventional model when all fruits in the image are targeted.

Figure 17 shows a filtered and averaged IoU of each fruit for each elapsed day. The filtered IoUs indicate the IoUs of the fruit recognized as the same fruit by the detection algorithm shown in Figure 5. The averaged IoU shows the mean value of the IoU of all fruits with the correct and overlapping regions. From Figure 17, the filtered IoUs for Fruit 1 and Fruit 2 are equivalent, but the averaged IoUs are lower for Fruit 1 than for Fruit 2. In Fruit 1, the averaged IoU is lower than the filtered IoU for most elapsed days due to overlap with other fruits. The averaged and filtered IoUs are similar in Fruit 2 as there is no overlap with other fruits. The results of Figure 17 show that the proposed method is able to track occluded fruit and accurately predict the region. Furthermore, the proposed fruit detection algorithm shown in Figure 5 enables accurate fruit detection and growth prediction even when the target fruit is occluded by other fruits and leaves, which suggests the effectiveness of the proposed fruit detection algorithm.

In addition, the information on the relative size change outputted by the proposed method is important, and if the actual fruit size is measured at any desired time, it is possible to easily graph the change over time and prediction results according to that size. For example, when the predicted results of this study were normalized and the harvesting fruit radius was assumed to be 4.5 cm based on the information from an apple sales website [26], all the predictions had an error of less than 0.5 cm after 195 elapsed days for both Fruit 1 and Fruit 2.

Furthermore, in actual management in an orchard using the proposed monitoring system, skilled farmers may be able to empirically identify trees that are representative of their farms and observations of a few representative canopies with a realistic number of fixed cameras will be considered. In addition, in an application of the proposed method in a mobile monitoring system, the composition of a mobile camera system changes, but the proposed method can cope with this by adjusting the detection range of the fruit tracking. The appropriate operation method will also be considered with regards to the size of the orchard and cost-effectiveness of the system.

## 4. Conclusions

In this study, an automatic training data generation method for hidden fruit detection and a fruit identification system were developed to predict the growth characteristics of individual apples in real time. To verify the effectiveness of the training data generation method, a transfer learning method was applied to the Mask R-CNN model using the automatically generated training data, and comparative experiments were conducted with the existing models. In addition, to verify the effectiveness of the individual identification system, individual fruit growth prediction was carried out on a series of farm images taken over some time.

The following conclusions can be derived from this study:The proposed model was capable of detecting hidden fruit with high accuracy, measuring fruit size in real time, and predicting the radius at harvest based on fruit size time series data.The model using automatically generated training data results was higher in precision and IoU compared to the existing model, which was affected by the parameters used to generate the synthetic farm images.In individual fruit identification, the same fruit could be identified at different dates and times by using filters for the amount of change in fruit position, size, and contour similarity.In real-time fruit radius measurements, the MAPE of the true and detection values was less than 0.079, and the coefficient of determination for linear regression was more than 0.95 for both Fruit 1 and 2, indicating that the fruit radius can be measured with high accuracy.In the real-time prediction of the radius at harvest, each prediction can capture a relative growth curve that is close to a true one after approximately 150 elapsed days.The proposed fruit detection algorithm enabled the tracking of even partially hidden fruit with sufficient prediction accuracy.

The present study enabled the detection of hidden fruit, which could not be detected in studies that conducted rectangular detection of fruit [6,7,8]. It also enabled individual fruit identification and fruit size measurements in real time. In addition, previously, annotating hidden regions required a lot of manual work and guesswork by the operator [21]. However, in this study, the cut-and-paste method was applied to automatically generate training data. The proposed method enables learning of the exact regions of the fruit, including the hidden regions, without the need for manual annotation.

In future studies, for more accurate fruit detection, the generation parameters of the training data will be adjusted and the boundary of the clipped image will be improved. We will also apply the model to fruits other than those verified in this experiment to confirm the generality of the model. Furthermore, in the fruit recognition accuracy of the proposed model, we will investigate a generation method that improves the recall value without reducing the precision and IoU by including smaller fruits in the training data. In addition, an algorithm that can accurately predict the radius at harvest earlier based on multi-year fruit growth curves will be developed. Moreover, the filter parameters in the fruit detection algorithm need to be adjusted according to the lens, distance to the subject and the condition of the fruit size and branch thickness, and their generalization will be discussed. In addition, the influence of the state of the hidden region and the variation in fruit size on the comparison of actual and predicted fruit size will be examined.

## Figures and Tables

**Figure 1 sensors-22-06473-f001:**
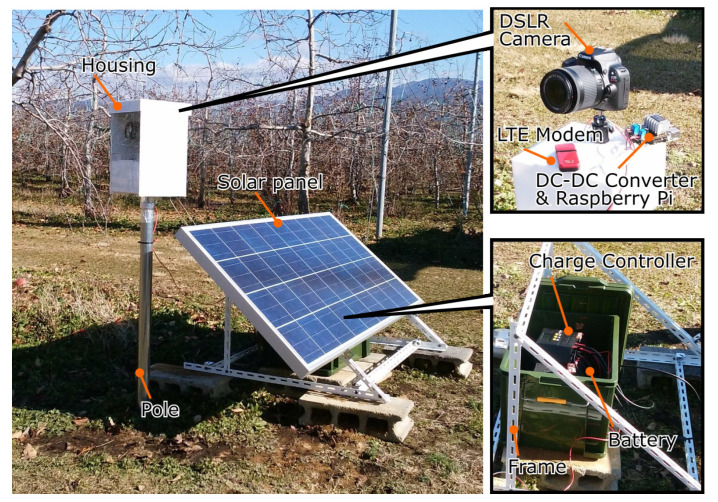
Monitoring device and its structure. The device sourced electric power from a solar panel and charged a 12 V battery via a charge controller. Two DC–DC converters, via a charge controller, converted electric power to 5 V and 7.4 V and supplied a Raspberry Pi 2 Model B microcontroller and DSLR camera. Images captured with the DSLR camera were temporarily stored on the microcontroller and then transmitted to a web server via a USB LTE modem.

**Figure 2 sensors-22-06473-f002:**
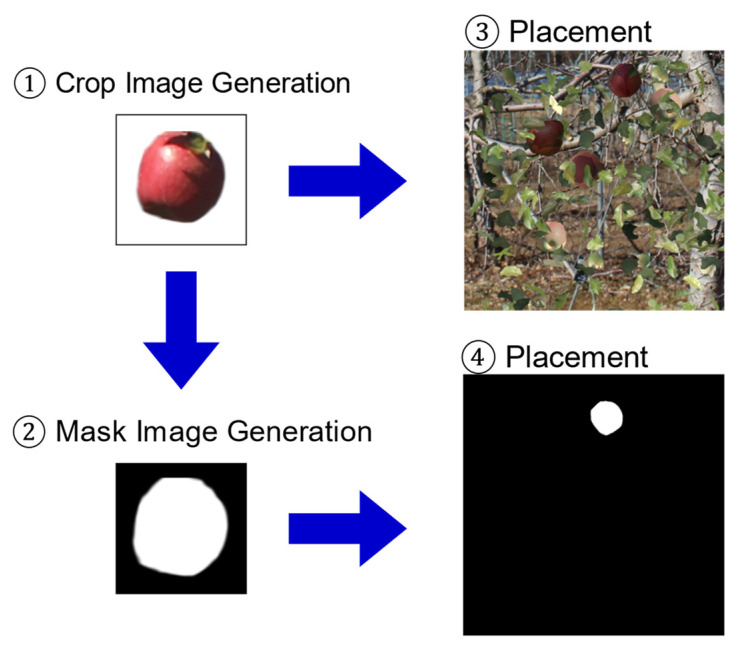
Generation process of synthetic images. First, fruit, leaf, and background images were manually cropped from actual farm images. Second, black-and-white mask images were created from the cropped fruit images to handle the shape information of the fruit. Third, fruit and leaf images were then pasted at random positions on the background image to generate a composite image of the canopy. Fourth, the generated fruit mask images were placed at the same position of the pasted fruit images on a black image, which was the same size as the background image.

**Figure 3 sensors-22-06473-f003:**
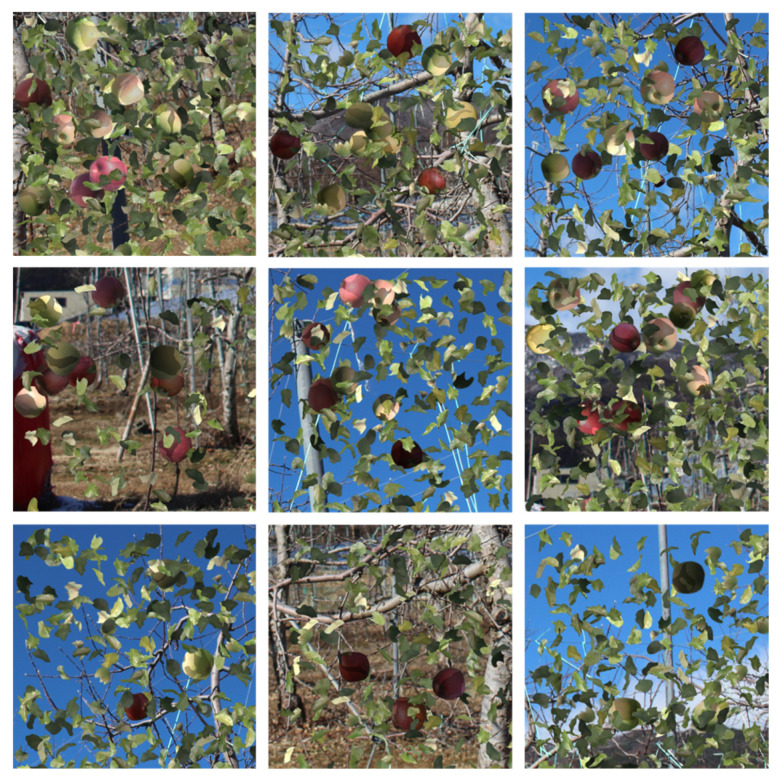
Examples of the generated training data. Red and green apples were randomly selected. Low luminance images were composited over the fruit image as a shadow of the leaf. For the shadow images, the aspect ratio, magnification, and angle were also varied to represent shadows of various shapes.

**Figure 4 sensors-22-06473-f004:**
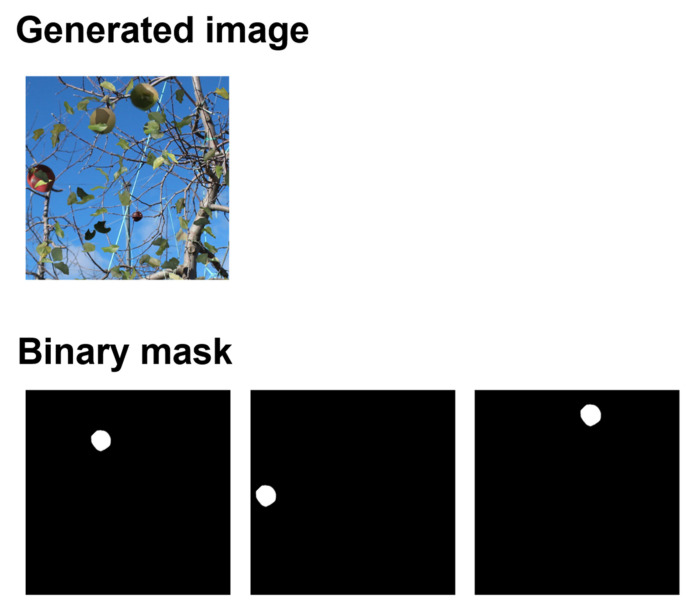
Binary mask and generated image. Each fruit is placed at the white position in the binary mask, respectively.

**Figure 5 sensors-22-06473-f005:**
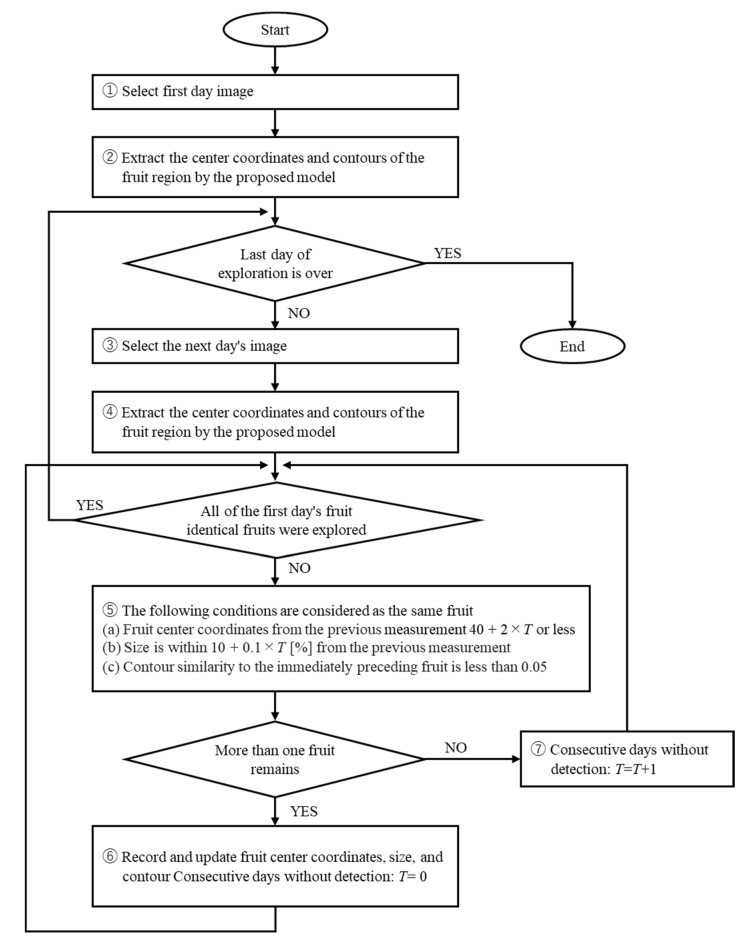
Flow chart of the fruit detection algorithm for tracking identical fruit in images captured at different dates and times on a farm. A three-layer filter was used to detect the same fruit at different times in part ⑤: a distance filter, size filter, and contour filter. The distance filter considered the hanging range of the fruit position caused by an increase in weight due to fruit growth; the size filter considered size variation due to growth and depth movement; the contour filter excluded fruits with incorrect contours and non-identical fruits with similar positions and sizes by calculating the contour similarity. *T* is the number of non-detect days, and it increases when the identical fruit is not found to mitigate the limitations of the distance and size filters in the search for identical fruit on the following day.

**Figure 6 sensors-22-06473-f006:**
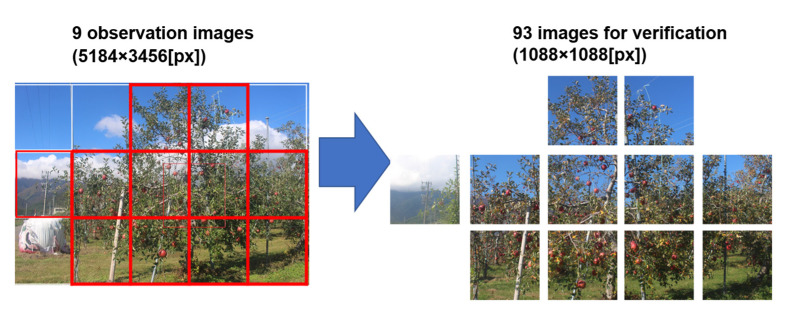
Generation of images for verification from observed images. Nine images (5184 × 3456 [px]), one for each day, were selected from the period 11 September to 20 November 2018, as images for validation data. Since the proposed model requires 1088 × 1088 [px] images as input data, each original image that is 5184 × 3456 [px] was cut and separated into 11 sections, generating 99 images with a resolution of 1088 × 1088 [px]. The 93 images were used as validation images because 6 of the images had no fruit.

**Figure 7 sensors-22-06473-f007:**
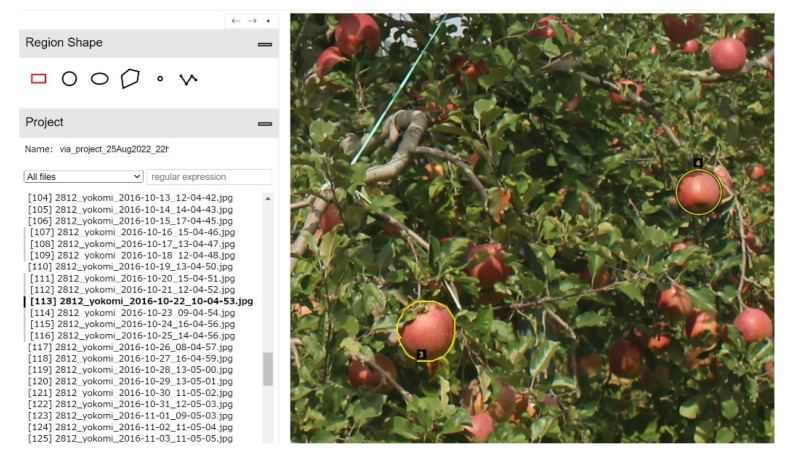
Example of annotation using VIA. Annotation of the regions was performed manually using the VGG Image Annotator, which could output the fruit contour in JSON format by enclosing the fruit region in the image.

**Figure 8 sensors-22-06473-f008:**
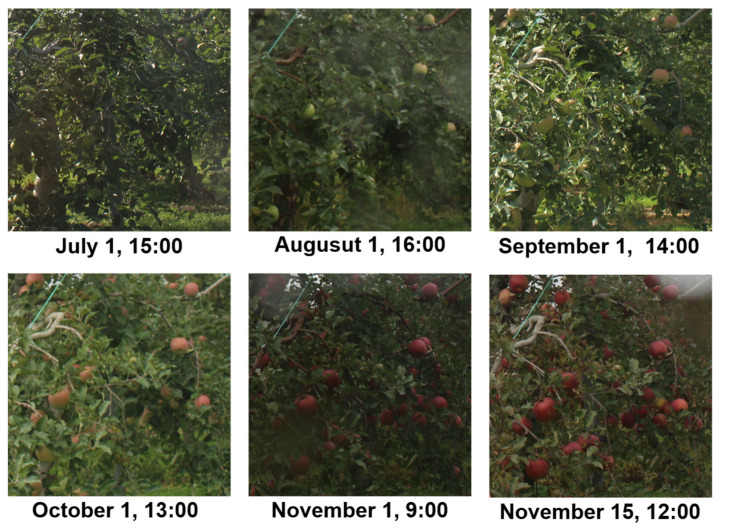
Examples of a validation image. One image from each day in which the fruit is clearly visible was selected from the 1417 farm images recorded over 137 days from 1 July to 15 November 2016.

**Figure 9 sensors-22-06473-f009:**
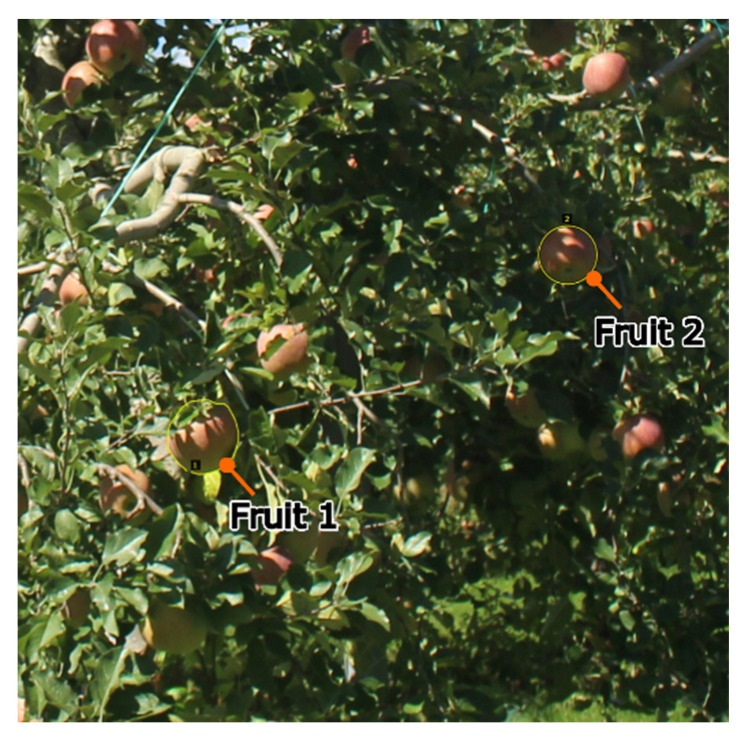
Target fruits to detect. Two fruit types (Fruit 1 and Fruit 2) in the farm image were targeted. Fruit 1 was partially hidden by leaves and other fruit in all images, while fruit 2 was partially hidden by leaves and other fruit in 23 of the 1417 images, which are several images per day ranging from 32 to 49 elapsed days.

**Figure 10 sensors-22-06473-f010:**
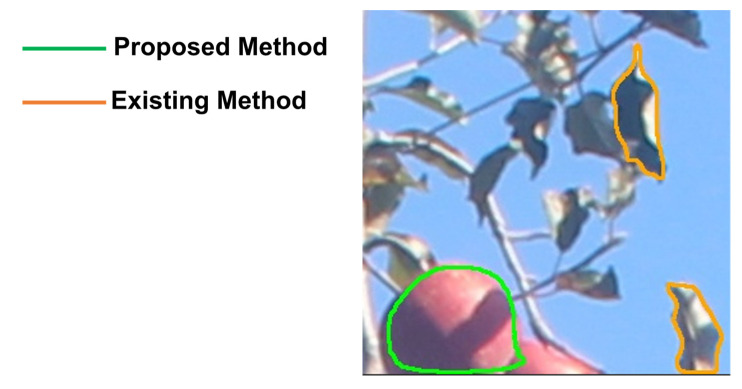
Example of a false detection. The orange line that is the result of the COCO learning model incorrectly detects leaves as fruit, while the green line that is the proposed model works correctly and does not incorrectly detect leaves.

**Figure 11 sensors-22-06473-f011:**
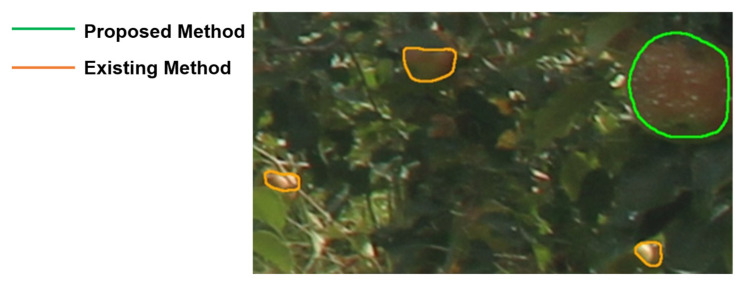
Example of small fruit detection. The proposed model outperformed the COCO learning model in mean and median IoU because it could accurately detect hidden regions of fruit without detecting parts of the fruit. The orange line is the result of the COCO as an existing method, and the green line is the result of the proposed method.

**Figure 12 sensors-22-06473-f012:**
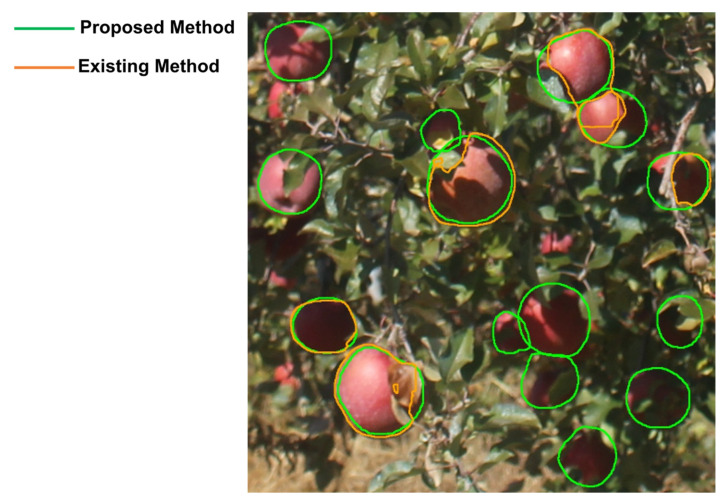
Example of hidden fruit detection. The proposed model can accurately detect hidden regions caused by fruit leaves, branches, and shadows. The orange line is the result of the COCO as an existing method, and the green line is the result of the proposed method.

**Figure 13 sensors-22-06473-f013:**
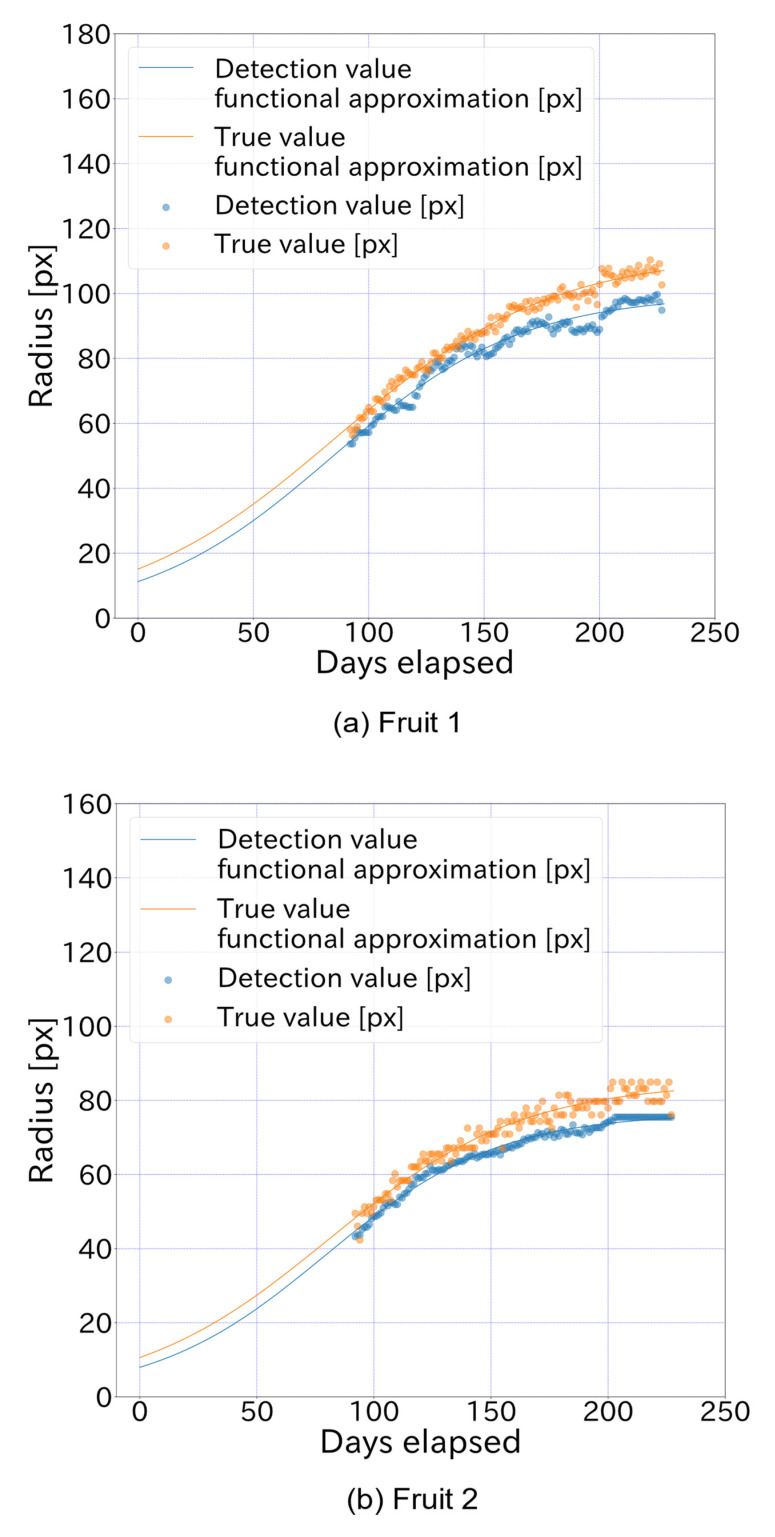
True radius values and detected radius values for each elapsed day. The growth curves derived from the true and detected values up to 227 elapsed days. In Fruit 1, a, b, and c in the functional approximation of Equation (4) are a = 99.71, b = 7.848, and c = −0.024 for the detection value curve and a = 112.24, b = 6.428, and c = −0.021 for the true value curve. In Fruit 2, a = 76.61, b = 8.595, and c = −0.027 for the detection value curve and a = 84.93, b = 7.001, and c = −0.024 for the true value curve.

**Figure 14 sensors-22-06473-f014:**
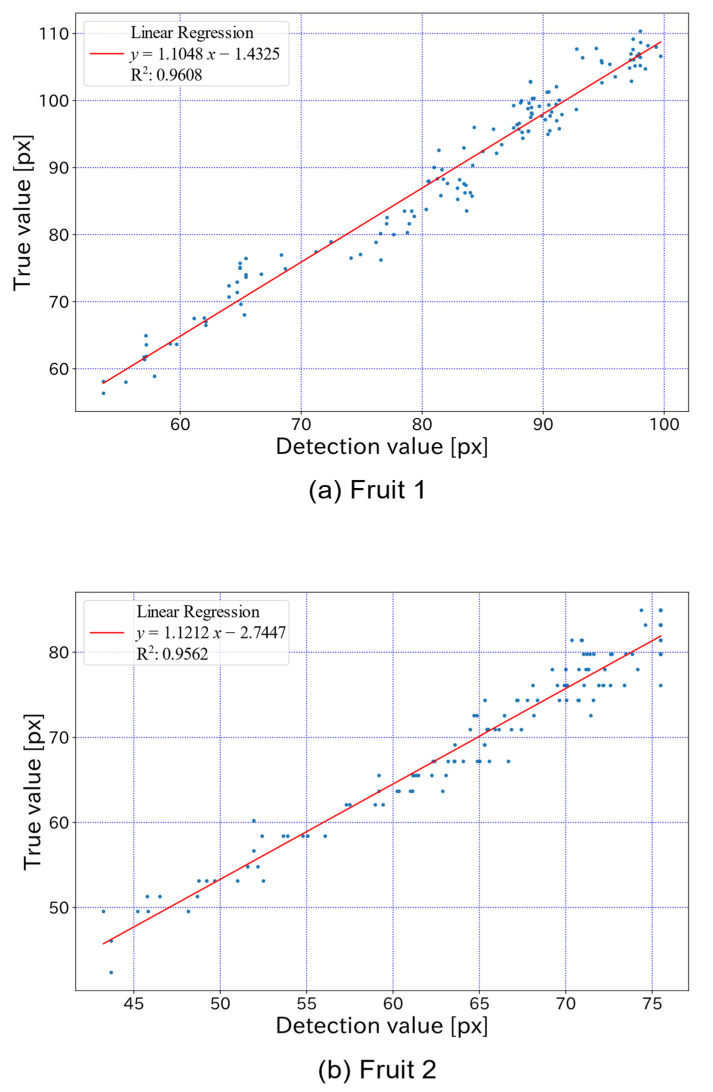
True values vs. detected values for the same elapsed days. The MAPE is 0.079 for Fruit 1 and 0.072 for Fruit 2, while the coefficient of determination for linear regression is approximately 0.96 for both Fruit 1 and Fruit 2.

**Figure 15 sensors-22-06473-f015:**
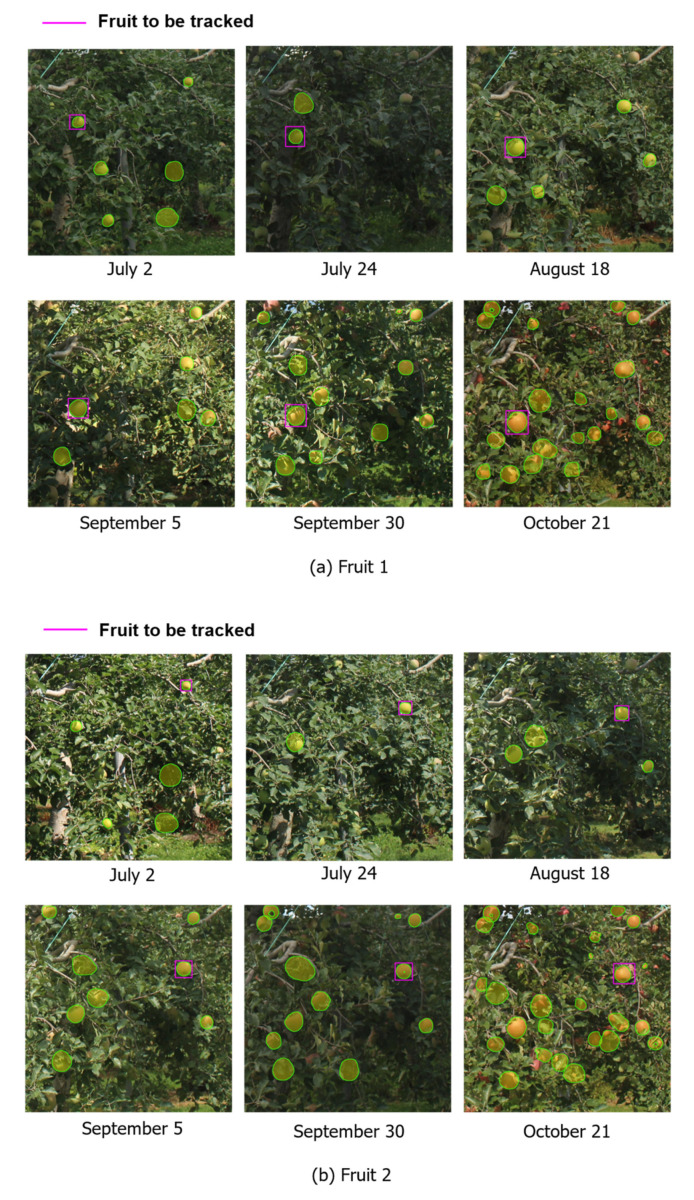
Examples of detections by the proposed model. Fruits detected are indicated by areas of green lines painted orange, and the fruit targeted for tracking are indicated by purple rectangles. Although there are some detection errors, the proposed method can detect individual fruits even when they are in contact with each other or densely packed together, as in the image from 21 October.

**Figure 16 sensors-22-06473-f016:**
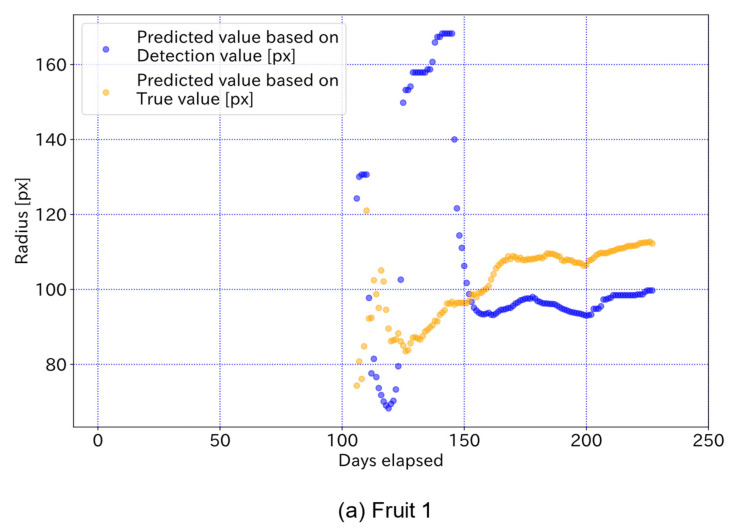
The fruit radius at harvest predicted from the true and detected values. Each prediction can capture the relative growth curve that is close to the true value after approximately 150 elapsed days, even if the target is partially hidden. Before 150 days, the prediction varies widely and has relatively large errors.

**Figure 17 sensors-22-06473-f017:**
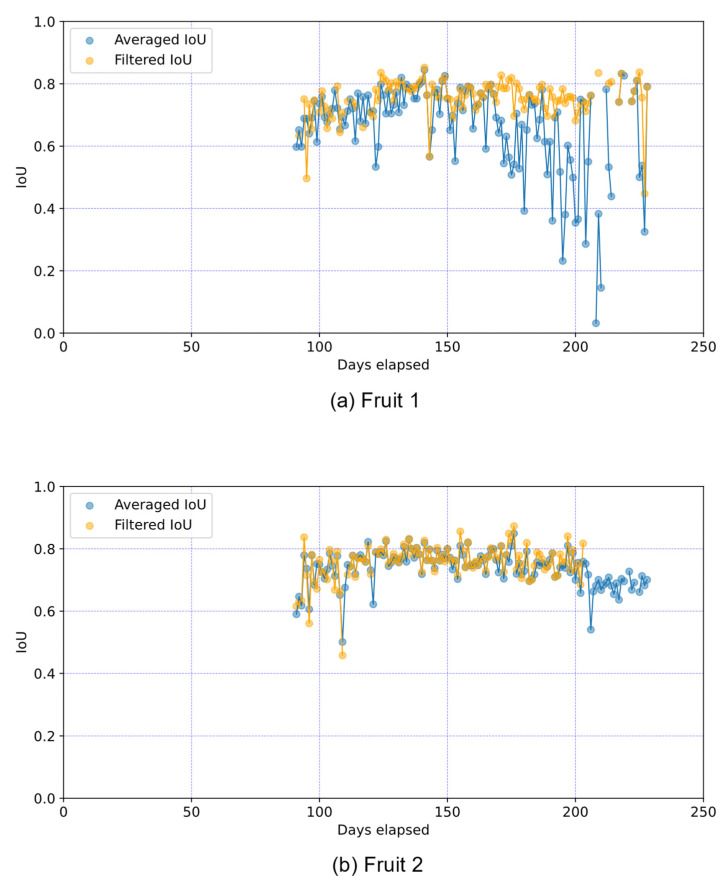
Filtered IoU and averaged IoU of each fruit for each elapsed day. The filtered IoUs indicate the IoUs of the fruit recognized as the same fruit by the proposed detection algorithm shown in Figure 5. The averaged IoU shows the mean value of the IoU of all fruits with the correct and overlapping regions. Although the averaged IoU of Fruit 1 is lower than that of Fruit 2 for most elapsed days due to overlaps with other fruits and leaves, the filtered IoUs of Fruit 1 and Fruit 2 are equivalent. The averaged and filtered IoUs are similar in Fruit 2 because there is little overlap with other fruits and leaves.

**Table 1 sensors-22-06473-t001:** Generation parameters. When placing the fruit and leaf images on a background image, the luminance, angle, magnification, and aspect ratio were randomly changed within an arbitrary range to increase the diversity of the training data.

Subject	Height [px]	Width [px]	Brightness [%]	Angle [°]	Number of Fruits	Number of Leaves on the Fruit Foreground	Number of Leaves on Fruit Background	Selection Probability of Red Apple [%]
Fruit	140~190	140~190	60~120	−90~90	3~8			55.3
Leaf	60~90	60~90	60~120	−90~90		80	0, 100, 200, 400	

**Table 2 sensors-22-06473-t002:** The validation results.

Item	Precision	Recall	IoU Average	IoU Median	IoU Variance
COCO model	0.864	0.338	0.554	0.596	0.065
Proposed model	0.955	0.317	0.653	0.720	0.046

## Data Availability

The data presented in this study are available on request from the corresponding author. The data are not publicly available due to privacy.

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
