# Peer review of "Real-Time Prediction of Growth Characteristics for Individual Fruits Using Deep Learning"

_sensors, 2022, doi:10.3390/s22176473_

Round 1
Reviewer 1 Report
The literature on fruit count by machine vision is growing rapidly, but that on on-tree fruit sizing is still nascent. Thus this topic area is appropriate for further consideration. For the journal Sensors the expectation should be to focus on the instrument (sensor) and the technique, rather than the application (agronomy).
There has, however, been more activity in the fruit sizing area than is credited in the Introduction, including reviews (eg https://www.mdpi.com/2073-4395/11/7/1409). I suggest this should be addressed. Conversely the very generic first paragraph of the Introduction could be deleted.
I feel the major novelty of the paper is in the claim to detect occluded fruit areas. I would like to see this area expanded, at the expense of other topics. The authors generate artificial images with clips of leaves, other fruit and shadows placed over fruit images in simulating canopy images. Thus the true fruit area is known, and used in training of the Mask-R-CNN model to predict location of the whole fruit, despite occlusion. Prediction is better than a traditional modelling procedure. This is an interesting result but deserves further study. Presumably the prediction of occluded regions of the fruit relies on an extrapolation based on the contour of the fruit, with the model ‘learning’ that the object (fruit) is spherical in this case. Could an edge-curve fitting procedure achieve a similar result? If too much (how much?) of the fruit contour is occluded the model must fail? How does the method work for non-spherical objects (fruit)? Li 236 speaks of the need to ‘set generation parameters appropriately in order to detect hidden regions’ – presumably this has something to do with curvature estimation – the reason behind this limitation should be further explored.
As written the paper focusses on use of machine vision for sizing fruit based on a static camera position. Fruit movement over time is accommodated, but this is not novel. The idea of sizing fruit on tree from a static camera position seems a bit odd from an application context: (i) the operator will need to set up the camera in exact locations each time; (ii) a relatively small number of fruit will be in view – is this representative of the orchard? How is this better than use of fruit dendrometers? Conversely others have used a population statistic approach, eg imaging of whole orchard blocks, with weekly estimate of average and SD of fruit size. (cf line 48 which claims tracking of individual fruit over time is required).
A 2D RGB image can only be used to size and object if camera to object distance is known (with application of the thin lens formulae). This can be achieved if a scale marker is present in the plane of the object or if object distance is known, eg from a ToF camera. Curiously this paper makes an assumption on fruit size from a reference web source (li 252) – this is odd.
Li 189-191 text meaning is unclear – ‘average value…was set as reference value, and the normalized detection and true values were defined as the magnification relative to the reference value’. As I haven’t understood this point I am lost on related discussion, eg li 244, Fig 12. What are the units on radius in fig 12?
The growth curves illustrate an application, but I think that for a Sensors article the focus should be on technique – in this case on the MaskRCNN estimation of size of occluded fruit. I would value a plot of predicted vs actual (as in measured with calipers) fruit size, with data collected across time providing variation in fruit size and states of occlusion.
Specific issues
Li 18 Abstract what is meant by ‘two types’ of fruit?
Li 41-42 how is fruit radius predicted from leaf area ratio?
Li 19 and elsewhere what is the unit on MSE? 0.0022 what?
Li 21 this seems odd – why need to assume actual radius is 4.5 cm
Li 84 please define camera resolution. What is image size per pixel at the camera to fruit distance used?
Li 93 use the term ‘canopy’ rather than ‘farm’
Fig 3 binary mask panels – how does this relate to the generated image?
Li150-155 reword to improve clarity
Li 158 how did the operator record hidden regions of fruit? A freehand annotation? What if they were allow to use a circle fitting tool?
Li 172 define a,b,c
Author Response
Dear Reviewer 1
Thank you very much for taking the time to review our manuscript in your busy schedule. We have responded to your question and made corrections and would be grateful if you would recheck the manuscript. The peer review response, revised manuscript and proofreading certificate on native checking of English are attached.
Best regards.
Yuya Aoyagi

Reviewer 2 Report
In this study, an automatic training data generation method for hidden fruit detection
and a fruit identification system were proposed to predict the growth characteristics of
individual apples in real-time. After verification, this method can extract the fruit size in real time and with high accuracy. This research can realize accurate growth management for the growth status of fruits, and is of great significance to improve the quality of products.
1. In lines 41-43, it is mentioned that genno et al. Predicted the average fruit radius relative to the whole image. Their research has certain similarities with the research in this paper. It is suggested to make a comparison with the research in this paper to highlight the differences or innovations of this paper.
2. In Table 1, there is a number "99" on the left side of "subject". If there is no special meaning, it needs to be deleted in the next modification.
3. For part ⑤ in Figure 4, does the same fruit need to meet three conditions at the same time or only one of them?
4. In lines 127-132, why do the distance filter, size filter, and contour filter extract these values? Are these values obtained by the author after long-term measurement? Or are there relevant references?
5. The references cited in line 176 should be [6]。
6. Too many paragraphs in the conclusion may cause some ambiguities. For example, lines 301-302 are placed here, which makes people feel abrupt and the meaning expressed is incomplete. Therefore, it is suggested to integrate the conclusion to make the meaning of the conclusion more accurate.
7. Also in the conclusion part, some contents are more suitable for the discussion part, such as lines 305-307. In addition, lines 308-313 are also more suitable for Results and discussion, so lines 308-313 can be integrated with lines 250-260.
8. In the late stage of apple growth (apple turns from green to red), adding the color and color changes of apple into the model may improve the detection accuracy of fruits in hidden areas. For example, the color of apples in the shadow or the color of apples covered by leaves in most areas may help improve the detection accuracy.
9. As shown in sections 2.2 and 3.1, the paper compares the accuracy of fruit recognition between the hidden fruit region detection model trained on the proposed training data and the model trained on the coco dataset, and the results show better results. However, the detection of apple size and the prediction of growth curve are also important research contents of this study. It is suggested to make a comparison between these parts and other related studies, so as to better reflect the improvement of this study.
Figure 12, Figure 13 and Figure 15 only provide the test results of two apples, which may reduce the persuasion of the conclusion obtained in this manuscript. If there are more test targets, the conclusion will be more persuasive.
Author Response
Dear Reviewer 2
Thank you very much for taking the time to review our manuscript in your busy schedule. We have responded to your question and made corrections and would be grateful if you would recheck the manuscript. The peer review response, revised manuscript and proofreading certificate on native checking of English are attached.
Best regards.
Yuya Aoyagi

Reviewer 3 Report
Manuscript is well-written but requires some major corrections. please consider the following comments:
- the research gap is lost in the abstract section
- in the introduction section, the novelty of the study is not well-justified
- dataset preparation section is lost
- choosing algorithm parameters are not justified. for example, choosing the type of filter or
- please add a subsection for presenting the precision and the resolution of the measuring instruments
- discussion section is weak
Author Response
Dear Reviewer 3
Thank you very much for taking the time to review our manuscript in your busy schedule. We have responded to your question and made corrections and would be grateful if you would recheck the manuscript. The peer review response, revised manuscript and proofreading certificate on native checking of English are attached.
Best regards.
Yuya Aoyagi

Round 2
Reviewer 1 Report
There are several area of value in the paper which deserve publication, but my opinion is that the revision has not done justice to these values. My recommendation is for a very major review, with attention to clarity of expression and potentially a split of the paper with addition of more material, to bring the ‘story’ into focus.
The two areas of interest are (i) the training of a model to predict area of occluded fruit and (ii) forecast of fruit size at harvest.
It is claimed that the apple is not spherical, that a Hough transform is inappropriate, and that synthetic data consisting of apple masks with leaf masks superimposed can be used to predict full apple area using a Mask RCNN. I am intrigued by these claims, but unclear on aspects of the work which deserve expansion for clarification.
However, I believe a clear presentation on prediction error on such synthetic data is essential, followed by presentation of error on fruit in actual canopy images. Also, while reference to another paper is made, I believe presentation on the measurement error associated with use of a circular outline compared to a freehand ‘guess’ at apple perimeter in occluded area is justified. Error should be presented in terms of number of pxels;
The use of normalization to the average fruit size over the first 15 days seems fraught, and confusing to the reader. In context of the claims being made, and focus to only 2 pieces of fruit, simple pixel number would be adequate (better yet would be a conversion to area, if camera to object distance was known). Certainly the use of normalized values does not add value in the Abstract – how can the reader interpret the Abstract assertion that “The mean squared error between the true value .. and detection value .. was less than 0.0022”? Also the failure to measure the actual size of the fruit, necessitating an assumption that radius was 4.5 cm is clumsy. Better to just present values in pixels.
The forward prediction of harvest size is a different topic, albeit founded on a machine vision technique. The current paper rests on a consideration of only two fruit – which appear not be occluded (though it is hard to see, given scale of images). The two sections of the paper thus seem ‘separable’.
The response to reviewers mentions “Therefore, the influence of the state of the hidden region and the variation in fruit size on the comparison of actual and predicted fruit size has not been fully investigated in this stage of our study.” – but how much stronger the paper would be if it had been investigated! Again, the use of values normalized to that of the average of the first 15 days of measurement seems unwarranted – presentation of pixel area values would be much more ‘direct’.
A response to reviewers mentions that a series of stationary cameras could be established across the orchard for harvest forecast use. How many fruit could be tracked in one image? How many cameras would be needed to capture a representative number of fruit per orchard? What would be the advantage of this compared to taking a mobile camera through the orchard, capturing a population size profile, at weekly intervals?
Li 203 where is reference to Table 4?
Author Response
Dear Reviewer 1
Thank you very much for taking the time to review our manuscript in your busy schedule. We have responded to your question and made corrections and would be grateful if you would recheck the manuscript. The peer review response and revised manuscript are attached.
Best regards.
Yuya Aoyagi

Reviewer 2 Report
The author solved all problem I concerned. I suggest the anthor could improve the manuscript according to other reviewers' comments.
Author Response

(The authors gave the same response as above.)
